# CineMorph: Learning Time-Continuous Motion Field for Motion Tracking on Cine Magnetic Resonance Images

## Abstract

Tracking cardiac motion using cine magnetic resonance imaging (cine MRI) is essential for evaluating cardiac function and diagnosing cardiovascular diseases. Current methods for cardiac motion tracking depend on scaling and squaring (SS) integration to learn discrete Lagrangian motion fields. However, this reliance hinders the effective exploitation of temporal continuity, leading to inadequate tracking accuracy. In this paper, we introduce a novel unsupervised learning method, CineMorph, to achieve temporally continuous cardiac motion tracking in cine MRI image sequences. Our approach integrates a frame-aware UNet with a series of time-continuous Transformer blocks to learn temporally continuous intra-frame motion fields, which are then assembled into time-continuous Lagrangian motion fields. To ensure the diffeomorphism property, we implement semigroup regularization to constrain our model, thus eliminating the reliance on SS integration. We evaluate our method on the public Automatic Cardiac Diagnostic Challenge (ACDC) dataset. The experimental results show that our method outperforms the existing state-of-the-art methods and achieves state-of-the-art performance with a mean DICE score of $83.6\%$ and a mean Hausdorff distance of $3.4$ mm.

## 1 Introdcution

Cine Magnetic Resonance Imaging (cine MRI) plays a crucial role in cardiac motion tracking due to its non-invasive nature and superior imaging capabilities Bello et al. (2019); Reindl et al. (2019); Wang et al. (2023). This technique allows for detailed visualization of the heart's anatomy and function throughout the cardiac cycle, capturing high-resolution images at multiple phases. By tracking the myocardial motion and deformation, clinicians can accurately assess cardiac function Sliman et al. (2014); Edvardsen et al. (2001), identify abnormalities in heart motion, and evaluate conditions such as myocardial infarction Reed et al. (2017), cardiomyopathies Ciarambino et al. (2021), and valvular diseases Coffey et al. (2021).

Compared to tagged MR images, cine MR images have the advantage of clearly visualizing cardiac anatomy, particularly the myocardium, as the epicardial and endocardial surfaces are distinctly visible. This makes it easier to track the radial motion of the myocardium. However, cine images fall short in accurately quantifying circumferential and longitudinal motion because there are few reliable features within the myocardium to track, and there are often insufficient long-axis images available Shi et al. (2012). Moreover, magnetic field inhomogeneities can cause variations in image brightness, especially with the balanced steady-state free precession (bSSFP) sequence, leading to dark band artifacts Ye et al. (2023).

In recent years, deep learning-based unsupervised methods have emerged as an efficient and effective design scheme for cardiac motion tracking Lu et al. (2023). These methods typically decompose the motion-tracking problem into pairwise registration processes. Using classical pairwise registration networks, such as VoxelMorph Balakrishnan et al. (2019), the motion field can be learned between two consecutive or any two images. When applied to consecutive images, the resulting motion fields need to be composed into Lagrangian motion fields to achieve motion tracking between any two images. The classical work is SequenceMorph Ye et al. (2023), which proposes a bi-directional generative diffeomorphic registration network to estimate the inter-frame motion field

between any two consecutive frames, and then recomposed them to the Lagrangian motion field between the reference frame and any other frame, through a differentiable composition layer. Considering temporal continuity between consecutive frames, SequenceMorph shows superior tracking performance and the feasibility of the motion decomposition and recomposition principle. Different from SequenceMorph, Lu *et al.* introduce the temporal relations and automatically learn spatiotemporal information from multiple images through a bidirectional recurrent neural network to directly estimate the Lagrangian motion field between the reference image and other images. However, these methods rely on the scaling and squaring integration scheme Hernandez et al. (2007); Arsigny et al. (2006) to reconstruct the deformation field. This reliance imposes a constraint on their capacity to capture temporal continuity, particularly for large deformation motions.

In this paper, we introduce a novel unsupervised learning method, called CineMorph, which generates time-continuous Lagrangian motion fields to facilitate smoother cardiac motion tracking. Drawing inspiration from Matinkia & Ray (2024), our method leverages the semigroup property Biagi & Bonfiglioli (2019) to learn the intra-frame motion field at any time and ensure diffeomorphic deformations without using scaling and squaring integration. To achieve this, we propose a new neural network architecture, which uses a frame-aware UNet Ronneberger et al. (2015) to encode two consecutive images with frame information and a series of transformer blocks to obtain time-continuous intra-frame motion fields. Benefitting from the time-continuous property, we further propose a time-continuous Lagrangian motion constraint to achieve global temporally-continuous motion tracking, as shown in Figure 1. To assess the effectiveness of our method, we conduct extensive experiments on the public ACDC dataset. Our results show our CineMorph is superior to the previous state-of-the-art models.

To sum up, our contributions can be summarized as the following:

- We introduce a novel unsupervised learning method for tracking cardiac motion in cine MRI images, which integrates a frame-aware UNet architecture with Transformer blocks to generate time-continuous Lagrangian motion fields.
- We propose a time-continuous Lagrangian motion constraint to ensure temporal continuity and diffeomorphism with semigroup regularization.
- We provide extensive experiments on the ACDC dataset, which demonstrate the superior performance of CineMorph over recent state-of-the-art methods.

## 2 RELATED WORK

**Optical Flow-Based Methods.** Optical flow (OF) is a widely used technique in video sequences to track objects by estimating the motion of objects between consecutive frames Brox & Malik (2010); Zhang et al. (2021); Xu et al. (2022); Shi et al. (2023); Saxena et al. (2024). OF can provide dense motion vectors for every pixel in the image, enabling detailed motion analysis across the entire frame. OF-based methods estimate cardiac motion field based on several basic assumptions regarding image appearance and motion strength, such as brightness consistency and small motion between the fixed and moving frames Carranza-Herrezuelo et al. (2010); Wang et al. (2019). However, these assumptions are not always valid in cardiac image sequences due to lighting changes, noise, or large displacements of the myocardium. Another challenge is that most OF-based methods require supervised learning, which is nearly impractical for medical images.

**Image Registration-Based Methods.** Image registration-based methods aim to find a transformation directly to obtain a dense displacement field that describes motion. Conventional non-rigid registration approaches, such as parametric B-Splines Rueckert et al. (1999), are formulated as iterative optimization procedures that maximize a similarity criterion between the fixed and moving images to determine the optimal transformation. Shi *et al.* developed a spatial and temporal registration approach that utilizes free-form deformations to estimate motion within the myocardium using a spatially-varying, weighted similarity measure Shi et al. (2012). Some studies have also utilized or extended this method to estimate cardiac motion for both untagged and tagged MR images Chandrashekara et al. (2004); De Craene et al. (2012). However, these methods are often associated with high computational costs and long execution times.

In recent years, there has been a surge of interest in applying deep learning to medical image registration and motion tracking Dalca et al. (2019); Niethammer et al. (2019); Chen et al. (2023). Compared

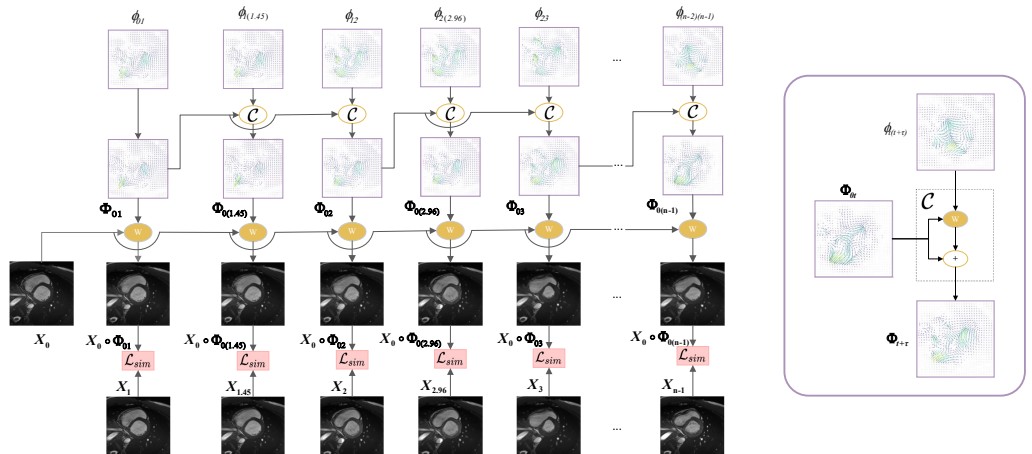

Figure 1: An overview of transforming motion field $\phi$ to Lagrangian motion field $\boldsymbol{\Phi}$ with a composition layer $\mathcal{C}$. As shown in the figure, our method can achieve temporally continuous motion tracking by estimating time-continuous Lagrangian motion fields. "w" means "warp".

to traditional iterative methods, deep learning-based approaches are faster and more accurate. In the context of motion tracking, the tracking problem is typically decomposed into pairwise registration processes to directly or indirectly generate Lagrangian motion fields Fechter & Baltas (2020); Yu et al. (2020) using registration networks Balakrishnan et al. (2019); Wu et al. (2022); Joshi & Hong (2023); Wang et al. (2024). Ye *et al.* proposed a bi-directional diffeomorphic registration network to estimate the inter-frame motion fields between consecutive image pairs and recompose them into Lagrangian motion fields through a differentiable composition layer Ye et al. (2021; 2023). Lu *et al.* proposed to model the temporal relations of cardiac cine MRI images through a bidirectional recurrent neural network to obtain the Lagrangian motion field between the reference image and other images Lu et al. (2023).

# 3 METHOD

## 3.1 PRELIMINARIES

### 3.1.1 MOTION DECOMPOSITION AND RECOMPOSITION

Cardiac cine MRI images capture a complete cardiac cycle, which comprises two phases: diastole and systole. Typically, the cine sequence starts at the end of diastole (ED), reaches peak contraction at the end of systole (ES), and then relaxes back to the ED phase. For a point $m$ in a cine image that moves from position $x_0$ at time $t_0$, we need to track its motion trajectory $x_t$. In an $N$-frame cine MRI image sequence, we only have the finite positions $x_n$ ($n = 0, 1, \cdots, N - 1$) of $m$. Over the time interval $\Delta t = t_{n-1} - t_{n-2}$, the displacement can be represented as a vector $\phi_{(n-2)(n-1)}$, also called inter-frame motion field. A sequence of such inter-frame motions $\{\phi_{t(t+1)}\}_{t=0}^{n-2}$ is composed to the Lagrangian motion field $\boldsymbol{\Phi}_{0(n-1)}$ Wang et al. (2019). Based on $\boldsymbol{\Phi}_{0(n-1)}$, we can shift the point $m$ from position $x_0$ to $x_{n-1}$. For motion tracking, given the first frame at time $t_0$ as the reference frame, our goal is to derive the Lagrangian motion field $\boldsymbol{\Phi}_{0(n-1)}$ between the reference frame and any subsequent frame at time $t_{n-1}$. Direct estimation of the Lagrangian motion field may lead to considerable motion errors due to large heart motion and intensity differences between temporally distant frames during the cardiac cycle. To address this, following Ye et al. (2023), we adopt the motion decomposition and recomposition principle, which first estimates the inter-frame motions $\{\phi_{t(t+1)}\}_{t=0}^{n-2}$ and then recomposes them to the Lagrangian motion field $\boldsymbol{\Phi}_{0(n-1)}$.

### 3.1.2 DIFFEOMORPHIC REGISTRATION FOR INTER-FRAME MOTION FIELD

For inter-frame motion field, deformable registration seeks for a vector field $\phi_{t(t+1)} : \mathbb{R}^2 \rightarrow \mathbb{R}^2$, which warps the moving image $\boldsymbol{X}_t$ at frame $t$ smoothly towards the fixed image $\boldsymbol{X}_{t+1}$ at frame

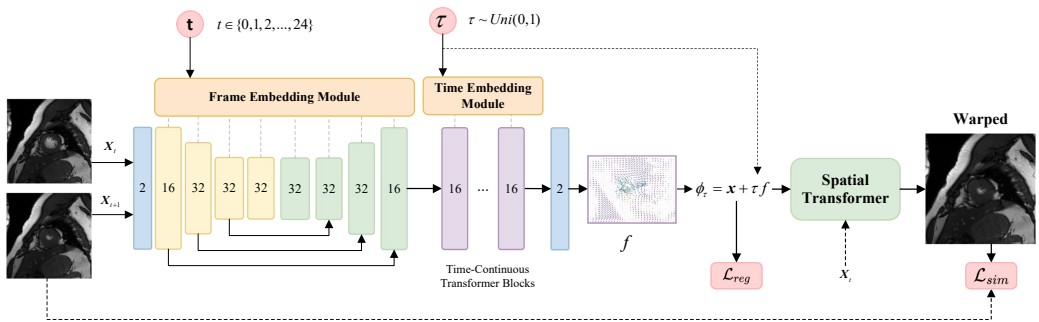

Figure 2: An overview of our proposed network. As illustrated in the figure, our frame-aware UNet is independent of time $\tau$. Therefore, when calculating the semigroup loss function, we only need to perform one forward propagation, reducing the training overhead.

$t + 1$. The deformation field $\phi$ is generally considered to be the flow map solution of the following ordinary differential equation (ODE) Beg et al. (2005); Chen et al. (2022); Joshi & Hong (2023); Wang et al. (2024):

$$\begin{cases} \frac{d\phi_\tau}{dt} = \mathbf{v}(\phi_\tau) = \mathbf{v} \circ \phi_\tau \\ \phi_0(\mathbf{x}) = \mathbf{x}, \end{cases} \tag{1}$$

where $\tau \in [0, 1]$, $\mathbf{x}$ is a spatial location, $\circ$ is a composition operator, $\mathbf{v}$ is a stationary velocity field and $\phi_0$ is an identity transformation. The utility of Equation (1) is that its solution is guaranteed to be a diffeomorphism:

$$\phi_{1/2^T} = \mathbf{x} + \frac{\mathbf{v}(\mathbf{x})}{2^T}. \tag{2}$$

$\phi_1$ can be obtained by using the scaling and squaring integration scheme with the recurrence $\phi_{1/2^i} = \phi_{1/2^{i+1}} \circ \phi_{1/2^{i+1}}$, which can be expressed as:

$$\phi_{1/2^{T-1}} = \phi_{1/2^T} \circ \phi_{1/2^T} \Rightarrow \cdots \Rightarrow \phi_1 = \phi_{1/2} \circ \phi_{1/2}. \tag{3}$$

A necessary and sufficient condition of $\phi$ as the flow map solution of Equation (1) is that it satisfies the semigroup property, i.e., for any time steps $\xi$ and $\varsigma$ it holds Biagi & Bonfiglioli (2019)

$$\phi_\xi \circ \phi_\varsigma = \phi_\varsigma \circ \phi_\xi = \phi_{\xi+\varsigma}. \tag{4}$$

Assuming that $\xi = -\varsigma$, we have $\phi_\xi \circ \phi_{-\xi} = \phi_0$ to guarantee the bijectivity of the deformation field $\phi$. Meanwhile, if the deformation $\phi$ satisfies Equation (4), then $\phi$ is a diffeomorphism at any time $\xi \in [-1, 1]$ Matinkia & Ray (2024).

## 3.2 PROPOSED METHOD

We propose an unsupervised deep learning method, dubbed as CineMorph, to learn a set of time-continuous motion fields $\{\phi_{t(t+\tau)}\}_{t=0}^{n-2}$, which are recomposed to the time-continuous Lagrangian motion fields. As shown in Figure 2, CineMorph consists of a frame-aware UNet and multiple time-continuous Transformer blocks. We decouple the frame $t$ and time $\tau$, allowing us to perform only a single forward propagation calculation with UNet when calculating the semigroup loss function. This reduces the computational cost and enhances the flexibility of the overall framework, as UNet can be substituted with other more sophisticated models.

### 3.2.1 FRAME-AWARE UNET

Considering the differences in myocardium motion across different frames, we introduce a frame-aware UNet that better models the motion features of the image pairs using a frame embedding module. The frame-aware UNet takes an image pair and frame $t$ as input and maps them to a motion feature. Formally, let $\boldsymbol{X}_t$ and $\boldsymbol{X}_{t+1}$ be a pair of 2D images with the same shape of $H \times W$ and let $\boldsymbol{Z} \in \mathbb{R}^{H \times W \times C}$ be the motion feature encoded by the frame-aware UNet $\psi$:

$$\boldsymbol{Z} = \psi(\boldsymbol{X}_t, \boldsymbol{X}_{t+1}, t; \boldsymbol{\theta}_1), \tag{5}$$

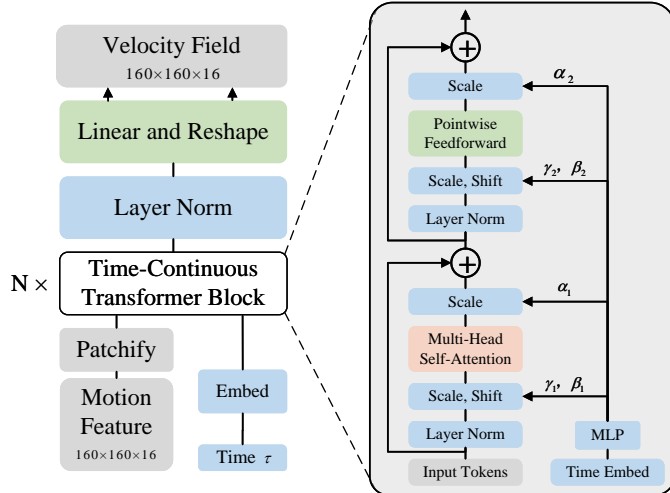

Figure 3: The architecture of time-continuous Transformer blocks. Left: The motion feature is decomposed into patches and processed by several transformer blocks. Right: Details of the time-continuous Transformer block.

where $\boldsymbol{\theta}_1$ represents the model parameters and $C$ represents the number of channels. The frame $t$ is encoded to an embedding vector of dimension $\mathbb{R}^d$ using a sinusoidal positional embedding $PE$ Vaswani (2017), followed by a multi-layer perceptron (MLP):

$$\boldsymbol{W}_2\sigma(\boldsymbol{W}_1PE(t)), \tag{6}$$

where $\boldsymbol{W}_1 \in \mathbb{R}^{d \times d}$ and $\boldsymbol{W}_2 \in \mathbb{R}^{d \times C}$ are learnable weights and $\sigma$ is the SiLU activation function. The embedding vector is added to UNet.

### 3.2.2 TIME-CONTINUOUS TRANSFORMER BLOCK

Inspired by Scalable Diffusion Transformers Peebles & Xie (2023), we propose to learn the time-continuous motion field $\phi$ using time-continuous transformer blocks. As illustrated in Figure 3, the time-continuous transformer block has a similar architecture to other transformer blocks Vaswani (2017). The key difference is that time $\tau$ is utilized as additional conditional information to regress the scale and shift parameters $\gamma$ and $\beta$, as well as the dimension-wise scaling parameters $\alpha$, through an MLP layer. The MLP is initialized to output the zero-vector for all $\alpha$, effectively setting the entire transformer block as the identity function. This ensures that the model focuses on learning inter-frame motion fields at the beginning of training. As training progresses, it gradually shifts to learning intra-frame motion fields.

**Patchify and Unpatchify.** The motion feature $\boldsymbol{Z}$ has a high spatial resolution ($160 \times 160$ in our experiment), significantly increasing the computational cost of the transformer blocks. Following Peebles & Xie (2023), we introduce a "patchify" layer as the first layer, which converts the motion feature $\boldsymbol{Z}$ into a sequence of tokens, each of dimension $d$, using a convolutional layer with kernel size $k$. After the final transformer block, we apply a final layer norm and linearly decode each sequence of image tokens. Finally, we rearrange the decoded tokens into their original spatial layout to obtain the predicted velocity field.

Different from Equation (2), we follow Matinkia & Ray (2024) to model the motion field $\phi$. Specifically, we construct a sequence of transformer blocks to map the motion feature $\boldsymbol{Z}$ to the motion field $\phi$:

$$\phi_\tau(\boldsymbol{x}, \boldsymbol{Z}; \boldsymbol{\theta}_2) = \boldsymbol{x} + \tau f(\boldsymbol{x}, \boldsymbol{Z}, \tau; \boldsymbol{\theta}_2), \forall \tau \in [-1, 1], \tag{7}$$

where $f$ is a sequence of transformer blocks with learnable parameters $\boldsymbol{\theta}_2$, which receives the motion feature $\boldsymbol{Z}$ rather than the pair of images. When $\tau = 0$, we have $\phi_0(\boldsymbol{x}) = \boldsymbol{x}$, hence satisfying the initial condition of the ODE 1. Additionally, to ensure that $\phi$ is a valid flow map, we enforce the

model to satisfy the semigroup property stated in Equation (4). We achieve this by setting $\xi = \tau$ and $\varsigma = \tau - 1$, which can be expressed as:

$$\phi_\tau \circ \phi_{\tau-1} = \phi_{\tau-1} \circ \phi_\tau = \phi_{2\tau-1}, \forall \tau \in [0, 1]. \tag{8}$$

By randomly sampling $\tau$, we can obtain the motion field $\phi_\tau$ at any time $\tau$, thus achieving the prediction of a continuous motion field.

### 3.2.3 INTER-FRAME AND INTRA-FRAME MOTION CONSTRAINTS

According to the bijectivity of the motion field, warping $\boldsymbol{X}_t$ up to time $\tau$ using the motion field $\phi_\tau$ must be equivalent to warping $\boldsymbol{X}_{t+1}$ up to time $1 - \tau$ using the inverse motion field $\phi_{\tau-1}$ due to the continuity of the trajectory of $\phi$. Hence we can define a time-continuous similarity loss:

$$\mathcal{L}_{sim}(\tau) = MSE(\phi_\tau[\boldsymbol{X}_t], \phi_{\tau-1}[\boldsymbol{X}_{t+1}]) = \|\phi_\tau[\boldsymbol{X}_t], \phi_{\tau-1}[\boldsymbol{X}_{t+1}]\|_2^2, \tag{9}$$

where $MSE$ is the mean squared error and $\phi_\tau[\boldsymbol{X}_t]$ represents warping $\boldsymbol{X}_t$ with $\phi_\tau$ using a spatial transformer network Jaderberg et al. (2015). $\mathcal{L}_{sim}$ measures inter-frame motion similarity when $\tau = 0$ or $\tau = 1$, and intra-frame motion similarity when $0 < \tau < 1$. The MSE loss is more suitable than normalized local cross-correlation (NCC) for image pairs that have similar intensity distributions and local contrast, such as cardiac Cine-MRI images Joshi & Hong (2023). Hence, we use the MSE loss in our experiments.

Using Equation 8, we impose the semigroup constraint on the motion field to ensure that $\phi$ is invertible and a diffeomorphism at all time steps:

$$\mathcal{L}_{reg}(\tau) = \|\phi_{2\tau-1} - \phi_\tau \circ \phi_{\tau-1}\|_2 + \|\phi_{2\tau-1} - \phi_{\tau-1} \circ \phi_\tau\|_2, \forall \tau \in [0, 1]. \tag{10}$$

We use an explicit smoothness to the motion field to ensure reasonable deformation by penalizing its gradients:

$$\mathcal{L}_{smooth}(\phi) = \|\nabla\phi\|_2^2. \tag{11}$$

Therefore, the inter-frame and intra-frame motion constraints are:

$$\mathcal{L}_1 = \mathbb{E}_{\tau \sim Uni(0,1)}[\lambda_0 \mathcal{L}_{sim}(\tau) + \lambda_1 \mathcal{L}_{reg}(\tau) + \lambda_2 \mathcal{L}_{smooth}(\phi)], \tag{12}$$

where $Uni(0, 1)$ is the uniform distribution on $[0, 1]$, and $\lambda_0$, $\lambda_1$ and $\lambda_2$ are the regularization factors.

### 3.2.4 TIME-CONTINUOUS LAGRANGIAN MOTION CONSTRAINTS

Benefiting from the prediction of the continuous motion fields $\{\phi_{t(t+\tau)}\}_{t=0}^{n-2}$, we can recompose them as time-continuous Lagrangian motion fields $\{\boldsymbol{\Phi}_{0(t+\tau))}\}_{t=0}^{n-2}$, with $\tau \in [0, 1]$, by a differentiable composition layer $\mathcal{C}$, as shown in Figure 1. Formally, we formulate the time-continuous Lagrangian motion fields as:

$$\boldsymbol{\Phi}_{0(t+\tau)} = \phi_{t(t+\tau)} \circ \boldsymbol{\Phi}_{0t}, \forall \tau \in [0, 1], \tag{13}$$

where $t = 0, 1, \cdots, N - 1$, $\boldsymbol{\Phi}_{00} = \phi_{00}$, and $\boldsymbol{\Phi}_{01} = \phi_{01}$.

With the Lagrangian motion field $\boldsymbol{\Phi}_{0(t+\tau)}$, we can warp the reference frame image $\boldsymbol{X}_0$ to any other time $t + \tau$: $\boldsymbol{\Phi}_{0(t+\tau)}[\boldsymbol{X}_0]$. By measuring the similarity between $\boldsymbol{X}_{t+\tau}$ and $\boldsymbol{\Phi}_{0(t+\tau)}[\boldsymbol{X}_0]$, we form a time-continuous Lagrangian motion consistency constraint:

$$\mathcal{L}_{lag}(\tau) = \frac{1}{N-1} \sum_{t=0}^{N-2} \mathcal{L}_{sim}(\boldsymbol{X}_{t+\tau}, \boldsymbol{\Phi}_{0(t+\tau)}[\boldsymbol{X}_0]), \tag{14}$$

where $N$ is the total frame number of a cine image sequence. $\tau$ follows a uniform distribution on $[0, 1]$. When $\tau = 0$ or $\tau = 1$, we use the ground truths $\boldsymbol{X}_t$ and $\boldsymbol{X}_{t+1}$ as labels to compute the loss $\mathcal{L}_{lag}$. Otherwise, we use $\boldsymbol{X}_{t+\tau} = \phi_\tau[\boldsymbol{X}_t]$ as a pseudo-label to to compute the loss $\mathcal{L}_{lag}$. Note that $\tau$ is independently sampled for each frame. Further, we also enforce the explicit smoothness of the Lagrangian motion field $\boldsymbol{\Phi}_{0(t+\tau)}$ by penalizing its gradients:

$$\mathcal{L}_{smooth}(\boldsymbol{\Phi}) = \|\nabla\boldsymbol{\Phi}\|_2^2. \tag{15}$$

The Lagrangian motion constraints are:

$$\mathcal{L}_2 = \mathbb{E}_{\tau \sim Uni(0,1)}[\lambda_3 \mathcal{L}_{lag}(\tau) + \lambda_4 \mathcal{L}_{smooth}(\mathbf{\Phi})], \qquad (16)$$

where $\lambda_3$ and $\lambda_4$ are the regularization factors to balance the contribution of each loss term. To sum up, the complete loss function $\mathcal{L}_{total}$ of our method is the sum of $\mathcal{L}_1$ and $\mathcal{L}_2$:

$$\mathcal{L}_{total} = \mathcal{L}_1 + \mathcal{L}_2. \qquad (17)$$

# 4 EXPERIMENTS

## 4.1 DATASET AND PRE-PROCESSING

We evaluated our method on the Automatic Cardiac Diagnostic Challenge (ACDC) dataset Bernard et al. (2018). ACDC is a public cine MR dataset that only consists of SAX view cine MR images from 150 subjects. Each scan includes 9 to 10 slices to cover the whole heart. In the original data split, there are 100 subjects in the training set, which includes segmentation mask annotations for the ED and ES frames, and another 50 subjects are in the testing set without any annotation masks. We rearranged and randomized the data based on subgroups, resulting in a revised configuration of 90 cases in the training set, 20 in the validation set, and 40 in the test set. We excluded slices located near the heart's base or apex due to the absence of annotation masks. The modified data contains 921 two-dimensional sequences in the training set, 180 in the validation set, and 388 in the test set, respectively. For each sequence, the number of frames varies from 12 to 35, covering only the ED to ES phases. If a sequence contains more than 25 frames, we removed extra frames from the sequence, except for the beginning and ending ones. Sequences with fewer than 25 frames remain unchanged. We first extracted the region of interest from the images to cover the heart, then resampled them to the same in-plane spatial size $160 \times 160$. Each sequence is used as input to the model for tracking the cyclic cardiac motion. Each input is a 2D sequence with a spatial resolution of $160 \times 160$ and a maximum of 25 frames. Following Ye et al. (2023), for each 2D image, we normalized the pixel values by first dividing them by 8 times the median intensity value of the image and then truncating the values to the range $[0, 1]$. Additionally, we performed data augmentation for each image with random rotation, translation, scaling, and Gaussian noise addition.

## 4.2 EVALUATION METRICS

We evaluated the motion tracking performance using the segmentation masks of the left ventricle (LV), myocardium wall (MYO), right ventricle (RV), and left atrium (LA). Since the mask annotations are available only on the ED and ES frames, we warped the mask from the ED frame to the ES frame using the estimated Lagrangian motion field. Here we used two metrics, the Dice score Dice (1945) and the $95\%$ maximum Hausdorff distance (HD95) Huttenlocher et al. (1993). The Dice score evaluates the degree of overlap between the estimated ES mask and the ground truth ES mask, while the HD95 measures the similarity of the region contours. A higher Dice and lower HD95 scores indicate better overlap between the two segmentation masks, reflecting superior tracking performance.

## 4.3 BASELINE METHODS

We compared our proposed method with three state-of-the-art methods: VoxelMorph (VM) Balakrishnan et al. (2018); Dalca et al. (2019), DeepTag Ye et al. (2021), and SequenceMorph (SM) Ye et al. (2023). For VM and DeepTag, we used their public implementations and retrained them from scratch, following the optimal hyper-parameters suggested by the authors. Since the code has not been released for SM, we report the results directly from their paper. We compare our method with SM without Lagrangian motion refinement (SM woR) for fair comparisons. VM is based on direct Lagrangian motion tracking, whereas DeepTag, SM, and our method are based on Lagrangian motion recomposition.

## 4.4 IMPLEMENTATION DETAILS

Our method was implemented with PyTorch. The architecture of the frame-aware UNet is similar to that described in Matinkia & Ray (2024). Specifically, the encoder has 3 down-sampling layers

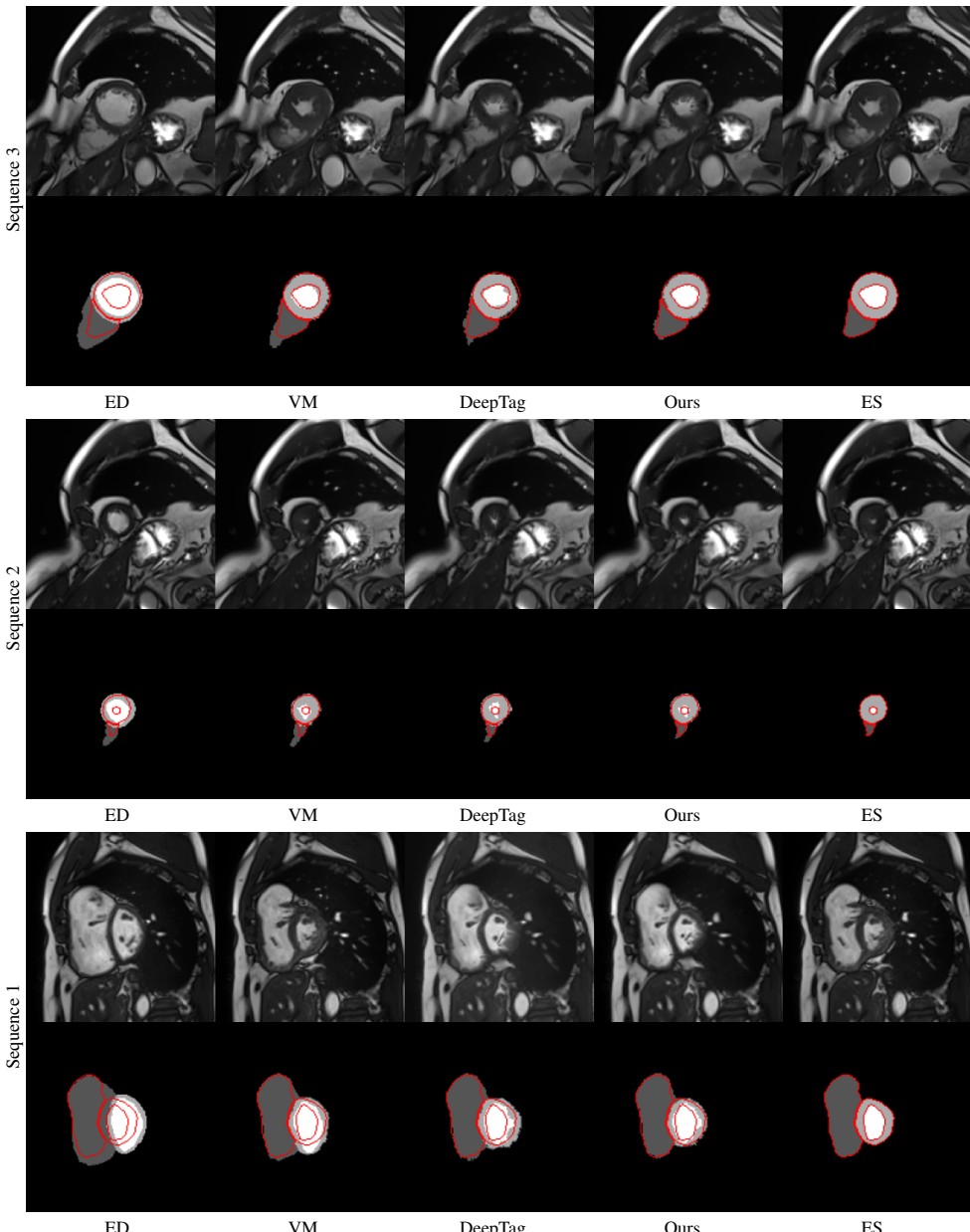

Figure 4: Motion tracking results on three cine MR image sequences (best viewed zoomed in). In each case, first row shows the images and second row shows the segmentation masks. Between ED and ES, we show the warped images by the estimated motion fields of different methods. Red contour shows the ground truth edge of LV, MYO and RV on the ES frame.

of dimensions 32, 32, and 32, and the decoder has 3 up-sampling layers with the same dimensions as the down-sampling layers. After the last up-sampling layer, we use a convolution layer to reduce the dimension to 16. All the activation functions for the layers are set to SiLU Hendrycks & Gimpel (2016) to provide more smoothness to the network. The number of time-continuous transformer blocks is set to 2. The time-embedding dimension is 64. The kernel size of the patchify layer is 8. We use the Adam optimizer with a $1e^{-4}$ learning rate to train our model for 1000 epochs. The regularization factors are set to $\lambda_0 = 100$, $\lambda_1 = 5e^8$, $\lambda_2 = 5$, $\lambda_3 = 50$, and $\lambda_5 = 1$, respectively.

## 4.5 RESULTS

**Motion tracking performance.** Table 1 provides a comprehensive comparison of the motion tracking performance of our method against other baseline methods. All values are expressed as mean

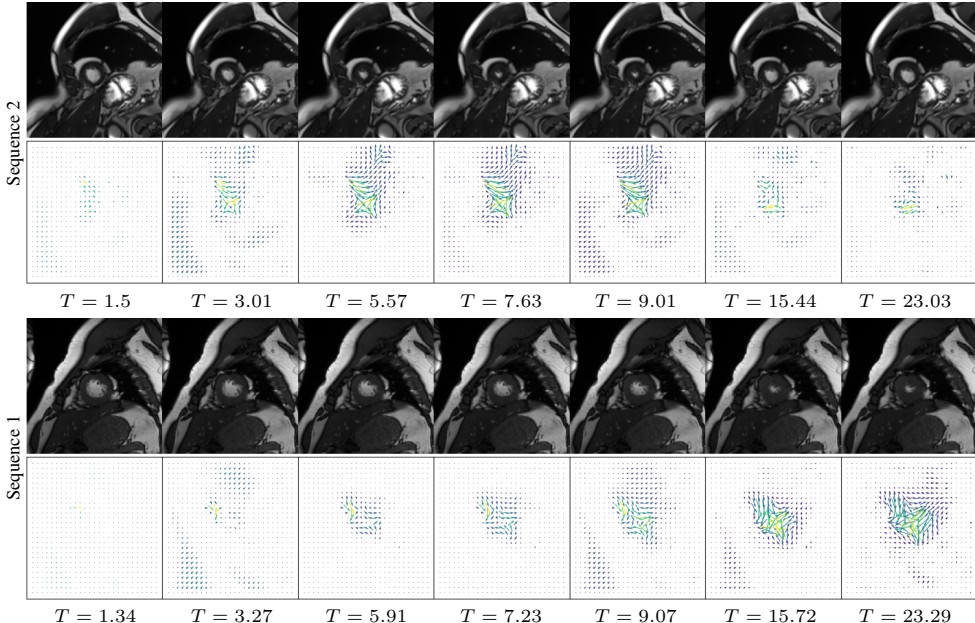

Figure 5: Motion tracking results on two cine MR image sequences (best viewed zoomed in). In each sequence, first row shows the warped images and second row shows the corresponding Lagrangian motion fields at different time $T = t + \tau$.

and standard deviation. Our implementation achieves similar motion tracking performance to that of Ye et al. (2023). As shown in Table 1, our method achieves the best performance regarding Dice and HD95 metrics. Compared to VM and DeepTag, our method consistently delivers better results for the LV, MYO, and RV regions. Compared to SM woR, our method shows significant performance improvements, except for the HD95 score in the MYO region. Figure 4 visualizes the warped images and motion tracking results of different methods from the ED phase to the ES phase on cine MR image sequences. The visualization shows that our method aligns more consistently with the ground truth of the ES mask. These results demonstrate the effectiveness of our method.

Table 1: Comparison of the performance of CineMorph with other methods. "woR" denotes "without Lagrangian motion refinement". "*" denotes that the results are reported in Ye et al. (2023).

| Method | Dice ↑ | | | | HD95(mm)↓ | | | |
|---|---|---|---|---|---|---|---|---|
| | LV | MYO | RV | avg | LV | MYO | RV | avg |
| VM* | $0.824 \pm 0.156$ | $0.793 \pm 0.105$ | $0.785 \pm 0.175$ | $0.801 \pm 0.021$ | $3.752 \pm 3.607$ | $3.071 \pm 2.399$ | $7.037 \pm 6.679$ | $4.620 \pm 2.121$ |
| VM (our impl.) | $0.827 \pm 0.170$ | $0.797 \pm 0.110$ | $0.765 \pm 0.208$ | $0.798 \pm 0.166$ | $3.657 \pm 2.508$ | $3.418 \pm 1.822$ | $5.484 \pm 3.731$ | $4.099 \pm 2.867$ |
| DeepTag* | $0.825 \pm 0.146$ | $0.793 \pm 0.094$ | $0.803 \pm 0.159$ | $0.807 \pm 0.016$ | $3.632 \pm 3.048$ | $2.924 \pm 1.819$ | $6.066 \pm 6.448$ | $4.208 \pm 1.648$ |
| DeepTag (our impl.) | $0.838 \pm 0.147$ | $0.796 \pm 0.093$ | $0.794 \pm 0.169$ | $0.810 \pm 0.139$ | $3.698 \pm 2.339$ | $3.501 \pm 1.672$ | $4.664 \pm 3.324$ | $3.907 \pm 2.523$ |
| SM woR* | $0.833 \pm 0.146$ | $0.802 \pm 0.094$ | $0.808 \pm 0.158$ | $0.815 \pm 0.017$ | $3.367 \pm 2.935$ | $2.787 \pm 1.808$ | $5.804 \pm 6.372$ | $4.016 \pm 1.652$ |
| Ours | $\mathbf{0.860 \pm 0.137}$ | $\mathbf{0.826 \pm 0.084}$ | $\mathbf{0.821 \pm 0.152}$ | $\mathbf{0.836 \pm 0.127}$ | $\mathbf{3.073 \pm 2.072}$ | $3.050 \pm 1.549$ | $\mathbf{4.081 \pm 3.273}$ | $\mathbf{3.356 \pm 2.384}$ |

**Visualization of the time-continuous Lagrangian motion field.** Benefiting from the prediction of the time-continuous Lagrangian motion fields, our method, compared to other tracking methods, can predict not only trajectories across frames but also intra-frame trajectories. By estimating the intra-frame motion field, our approach makes the motion field smoother, thereby improving tracking performance. In Figure 5, we visualize the warped images and corresponding Lagrangian motion fields at different time.

### 4.6 Ablation Study

**Effects of time-continuous transformer blocks.** To investigate the impact of time-continuous transformer blocks on model performance, we train our model with varying numbers of blocks. Considering when the number of the transformer block is 0, the semigroup property is not used to constrain our model. In this case, we change the input of the frame embedding module to the time $\tau$ sampled from $Uni(0, 1)$. The results are reported in Table 2. We find that the transformer block yields considerable performance improvement, indicating the transformer block is critical to

improving motion tracking performance. Again, we observe that across different configurations, similar average Dice and HD95 scores are obtained by increasing the number of blocks, indicating that our method is insensitive to the number of transformer blocks. However, further increasing the number of blocks will increase the computational cost. Therefore, in our experiments, we set the number of blocks to 2 by default.

Table 2: Results of our method with varying numbers of transformer blocks.

| Number | Dice ↑ | | | | HD95(mm)↓ | | | |
|---|---|---|---|---|---|---|---|---|
| | LV | MYO | RV | avg | LV | MYO | RV | avg |
| 0 | $0.848 \pm 0.145$ | $0.818 \pm 0.092$ | $0.815 \pm 0.158$ | $0.828 \pm 0.134$ | $3.331 \pm 2.130$ | $3.147 \pm 1.574$ | $4.212 \pm 3.284$ | $3.520 \pm 2.409$ |
| 1 | $0.859 \pm 0.140$ | $0.828 \pm 0.085$ | $0.817 \pm 0.157$ | $0.836 \pm 0.130$ | $3.060 \pm 2.066$ | $2.987 \pm 1.542$ | $4.146 \pm 3.291$ | $3.348 \pm 2.397$ |
| 2 | $0.860 \pm 0.137$ | $0.826 \pm 0.084$ | $0.821 \pm 0.152$ | $0.836 \pm 0.127$ | $3.073 \pm 2.072$ | $3.050 \pm 1.549$ | $4.081 \pm 3.273$ | $3.356 \pm 2.384$ |
| 3 | $0.858 \pm 0.141$ | $0.828 \pm 0.083$ | $0.817 \pm 0.159$ | $0.835 \pm 0.131$ | $3.069 \pm 2.127$ | $3.040 \pm 1.562$ | $4.130 \pm 3.304$ | $3.366 \pm 2.421$ |

**Importance of time-continuous Lagrangian motion constraint.** To evaluate the effectiveness of our proposed time-continuous Lagrangian motion constraint (TCLMC), we train our model with and without TCLMC. Table 3 shows the motion tracking performance. Our proposed TCLMC significantly improves the model's performance in terms of Dice and HD95 scores. TCLMC helps the model learn more continuous motion fields and reduces the drift error accumulating over time, resulting in better motion estimation on a series of frames.

Table 3: Ablation study on the time-continuous Lagrangian motion constraint (TCLMC).

| TCLMC | Dice ↑ | | | | HD95(mm)↓ | | | |
|---|---|---|---|---|---|---|---|---|
| | LV | MYO | RV | avg | LV | MYO | RV | avg |
| × | $0.848 \pm 0.150$ | $0.815 \pm 0.094$ | $0.811 \pm 0.160$ | $0.826 \pm 0.137$ | $3.270 \pm 2.135$ | $3.117 \pm 1.575$ | $4.209 \pm 3.326$ | $3.487 \pm 2.431$ |
| √ | $\mathbf{0.860 \pm 0.137}$ | $\mathbf{0.826 \pm 0.084}$ | $\mathbf{0.821 \pm 0.152}$ | $\mathbf{0.836 \pm 0.127}$ | $\mathbf{3.073 \pm 2.072}$ | $\mathbf{3.050 \pm 1.549}$ | $\mathbf{4.081 \pm 3.273}$ | $\mathbf{3.356 \pm 2.384}$ |

**Effects of frame embedding module.** Here we study the effects of the frame embedding module with three embedding ways. First, we remove the frame embedding module to explore its importance for motion tracking (Model A). Second, we replace the frame $t$ with the time $\tau$ as the input to the frame embedding module (Model B). Third, we maintain the frame embedding module (Model C). The results are shown in Table 4. We find that Model B is superior to Model A, demonstrating the effectiveness of the frame embedding module. Models A and B achieve similar Dice and HD95 scores, indicating that the frame $t$ and the time $\tau$ are interchangeable. However, the advantage of using the frame $t$ as the input is that when implementing the semigroup property, only one forward propagation of the UNet is required, whereas using the time $\tau$ requires three propagation, significantly reducing computation costs. Additionally, we can use more complex models, such as TransMorph Chen et al. (2022), to train our model for more accurate motion tracking.

Table 4: Ablation study on the frame embedding module. A: without the frame embedding module. B: Replacing the frame $t$ with the time $\tau$ as the input to the frame embedding module. C: with the frame embedding module.

| Model | Dice ↑ | | | | HD95(mm)↓ | | | |
|---|---|---|---|---|---|---|---|---|
| | LV | MYO | RV | avg | LV | MYO | RV | avg |
| A | $0.854 \pm 0.143$ | $0.825 \pm 0.091$ | $0.818 \pm 0.157$ | $0.833 \pm 0.132$ | $3.215 \pm 2.165$ | $3.079 \pm 1.584$ | $4.175 \pm 3.323$ | $3.444 \pm 2.443$ |
| B | $0.859 \pm 0.139$ | $0.827 \pm 0.088$ | $0.821 \pm 0.151$ | $0.836 \pm 0.128$ | $3.123 \pm 2.088$ | $3.064 \pm 1.599$ | $4.089 \pm 3.291$ | $3.381 \pm 2.406$ |
| C | $0.860 \pm 0.137$ | $0.826 \pm 0.084$ | $0.821 \pm 0.152$ | $0.836 \pm 0.127$ | $3.073 \pm 2.072$ | $3.050 \pm 1.549$ | $4.081 \pm 3.273$ | $3.356 \pm 2.384$ |

## 5 CONCLUSION

In this paper, we present a novel unsupervised learning method for generating time-continuous Lagrangian motion fields to improve cardiac motion tracking in cine MRI images. Our approach utilizes a frame-aware UNet to encode two consecutive images with frame information and employs a series of transformer blocks to derive time-continuous intra-frame motion fields. We train our model using semigroup regularization and time-continuous Lagrangian motion regularization to capture temporal continuity and ensure diffeomorphism. Extensive experiments on the public ACDC dataset demonstrate the effectiveness of our method.

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
