# OpenReview forum: "CineMorph: Learning Time-Continuous Motion Field for Motion Tracking on Cine Magnetic Resonance Images"
_ICLR.cc/2025/Conference — Submitted to ICLR 2025_

### Official Review · Reviewer_T6Nm · 2024-10-28

**Soundness:** 1
**Presentation:** 2
**Contribution:** 1
**Rating:** 3
**Confidence:** 5

**Summary:**

Add a Transformer block named Time-Continuous Transformer block in the DeepTag framework [1] for motion tracking

[1] Ye, M., Kanski, M., Yang, D., Chang, Q., Yan, Z., Huang, Q., Axel, L. and Metaxas, D., 2021. Deeptag: An unsupervised deep learning method for motion tracking on cardiac tagging magnetic resonance images. In Proceedings of the IEEE/CVF conference on computer vision and pattern recognition (pp. 7261-7271).

**Strengths:**

Add a Transformer block named Time-Continuous Transformer block in the DeepTag framework. This block is used to formulate the temporal information of cardiac motion.

**Weaknesses:**

1. Except for the 3.2.2 method, all other parts can be found in other papers. This means the original part is only the transformer block.

2. This paper introduces the Time-Continuous Transformer block, which can be considered as adding the temporal positional encoding to handle the motion relationship. However, such methods have already been explored in action recognition tasks by using the Transformer architecture (ViViT [2], Swim-Transformer [3], TimeSformer [4], VideoLightFormer [5], MViT [6], etc.). All these works serve for temporal feature extraction, and they also have several ways to handle the temporal positional embedding or add the learnable temporal feature embedding as prior knowledge. This paper has simply adapted such a method in motion tracking while not proposing any original creative/innovation.

3. The experiment is not enough; the cardiac motion can not only be seen/detected by cine MRI, but echocardiography also can be used to visualize the cardiac motion. SequenceMorph also conducted the experiment using the echocardiography dataset (CAMUS dataset). Can this method also be efficiently applied to echocardiography datasets?

4. The survey related to temporal formulation approaches of this paper is not sufficient. In this paper, the temporal information (formulated by $\tau$) can be treated as latent coding. Currently, there exist many methods/approaches that can integrate such latent code. Why only use the Transformer block? In my opinion, the diffusion-based approach can also be adapted to the motion tracking/registration tasks. The temporal information ($\tau$) can also be decoded or encoded as the latent coding for diffusion methods.

5. What's the efficiency experiment? As you add extract blocks/parameters into the network, the computational cost will also increase. Can computational cost remain almost the same or only slightly improve when equipping the Transformer blocks? Also, only a few methods have been compared in this paper. In recent years, many works that focus on motion-tracking have been proposed; the insufficient survey of motion-tracking methods is also a drawback of this paper.

[2] Bertasius, G., Wang, H. and Torresani, L., 2021, July. Is space-time attention all you need for video understanding?. In ICML (Vol. 2, No. 3, p. 4).

[3] Arnab, A., Dehghani, M., Heigold, G., Sun, C., Lučić, M. and Schmid, C., 2021. Vivit: A video vision transformer. In Proceedings of the IEEE/CVF international conference on computer vision (pp. 6836-6846).

[4] Liu, Z., Ning, J., Cao, Y., Wei, Y., Zhang, Z., Lin, S. and Hu, H., 2022. Video swin transformer. In Proceedings of the IEEE/CVF conference on computer vision and pattern recognition (pp. 3202-3211).

[5] Koot, R. and Lu, H., 2021. Videolightformer: Lightweight action recognition using transformers. arXiv preprint arXiv:2107.00451.

[6] Fan, H., Xiong, B., Mangalam, K., Li, Y., Yan, Z., Malik, J. and Feichtenhofer, C., 2021. Multiscale vision transformers. In Proceedings of the IEEE/CVF international conference on computer vision (pp. 6824-6835).

[7] Ye, M., Yang, D., Huang, Q., Kanski, M., Axel, L. and Metaxas, D.N., 2023. SequenceMorph: A unified unsupervised learning framework for motion tracking on cardiac image sequences. IEEE Transactions on Pattern Analysis and Machine Intelligence, 45(8), pp.10409-10426.

**Questions:**

Please see the weaknesses.

---

### Official Review · Reviewer_CLVf · 2024-11-02

**Soundness:** 3
**Presentation:** 3
**Contribution:** 2
**Rating:** 5
**Confidence:** 3

**Summary:**

The authors present an unsupervised learning approach for continuous motion tracking in cine MRI sequences, which is a well studied problem. The method leveraged frame-aware UNet, and introduced a time continuous Transformer blocks. The authors proposed a time-continuous Lagrangian motion constraint to ensure temporal continuity and diffeomorphism with semigroup regularization. The results on ACDC dataset demonstrates its effectiveness.

**Strengths:**

Originality: The authors introduced a few novel techniques to improve the tracking of cine cardiac motion on 2D SAX view, including the new regularization to ensure time continuous Lagrange motion field.

Clear presentation and comprehensive ablation study on the proposed approaches.

**Weaknesses:**

The proposed approach is presented as a specialized method for cine cardiac motion but offers only marginal improvements. The authors did not attempt to validate the method on different datasets or in other medical imaging problems—such as cardiac ultrasound, cardiac imaging in the long-axis view, etc., which severely limits its appeal to the ICLR audience. It may be more suitable for submission to a medical imaging conference (e.g., MICCAI). Ideally, the proposed method should not be limited to cardiac motion but applicable to any smooth motion or deformation in anatomical structures. Conducting experiments on a limited dataset reduces its perceived applicability.

While different evaluation sets were identified, the state-of-the-art performance on the ACDC dataset appears better than the results reported in this paper. For example, Yu et al.'s "Motion Pyramid Networks for Accurate and Efficient Cardiac Motion Estimation" (2020) reported tracking performance with a Dice score of 0.9x, evaluated on multiple datasets.

Since the proposed method can be applied to any frame between end-diastole (ED) and end-systole (ES), it would be very interesting to see how the difference map looks when the method is used to wrap images with the derived motion fields for the whole cardiac cycle from the ED frame to the ES frame and played as a movie. Qualitatively assessing the myocardium motion compared to the original cine MRI sequence would add significant value.

The paper lacks a discussion of the computational overhead of the proposed features. It would be beneficial for readers to understand the impact on training and inference speed, especially since the improvements over other methods do not seem that substantial.

Figure explanations can be further improved. It took considerable time to understand Figures 1 and 2, moving back and forth between the text and the visuals. Enhanced explanations would be appreciated by the readers.

Overall, the manuscript's contributions to cardiac motion tracking in cine MRI seem to be modest. The presented results, while somewhat improved, may not sufficiently establish CineMorph as a new standard in the field. Expanding on the method's computational implications and applicability to diverse imaging contexts could help strengthen the paper's overall impact.

**Questions:**

In Section 4.6, the first experiment shows that further increases in time-continuous transformer blocks seem to decrease performance. Is there any insight into why this occurs? I can understand choosing 2 instead of 3 for computational reason but cannot understand why 3 decreases the performance. Also, according to Table 2, using one block appears to achieve the best performance when considering both Dice and HD95 metrics. Why did you choose to use 2 blocks instead of 1? (Choosing 1 can further reduce computational overhead)

As the authors mentioned that CineMorph performs similar to SM with Lagrange motion refinement, does it worth to literally compare these two methods at least.

The method is an unsupervised learning method for motion estimation but the evaluation is on segmentation. How could we know the motion fields generated by the proposed method is the better one esp given similar performance on segmentation metrics. (This is not a problem only for this paper but would like to get authors' thoughts on this)

---

### Official Review · Reviewer_bxEt · 2024-11-03

**Soundness:** 1
**Presentation:** 2
**Contribution:** 2
**Rating:** 3
**Confidence:** 5

**Summary:**

This paper presents a multi-frame cardiac motion estimation method, integrating transformer blocks within a learning-based, unsupervised image registration framework.

**Strengths:**

Cardiac motion estimation is a complex and valuable problem. The authors’ approach shows potential for enabling continuous motion estimation throughout the cardiac cycle, marking a promising advancement in this area.

**Weaknesses:**

1. The estimated displacement in Figure 5 appears implausible, with arrows exhibiting non-smooth trajectories and incorrect directions. For example, during the contraction phase, displacement vectors should predominantly point toward the left ventricle, as demonstrated in Fig. 4 of [Reference: https://arxiv.org/abs/2312.00837]. The final motion fields in sequences 1 and 2 (Figure 5) lack realism.

2. The approach is broadly applicable to motion estimation; however, testing exclusively on the ACDC dataset is insufficient to evaluate its generalizability and reproducibility.

3. The method does not address variable frame rates during inference, and it appears to require a consistent number of input frames, limiting flexibility across different cardiac sequences.

**Questions:**

1. Have the authors quantitatively assessed the physical plausibility of the estimated displacement fields? For instance, evaluating the negative Jacobian determinant or similar metrics could be informative.

2. The evaluation primarily covers ED-to-ES registration, which may not fully capture the quality of intermediate motion fields. Testing on datasets with intermediate ground-truth labels would better validate the estimated motion fields’ accuracy.

---

### Comment · Area_Chair_QxBU · 2024-11-27
**Reminder: Last day for author feedback**

This is a reminder that today is the last day allotted for author feedback. If there are any more last minute comments, please send them by today.

---

### Meta-Review · Area_Chair_QxBU · 2024-12-19

**Metareview:**

The authors proposed an approach for cardiac motion using cine magnetic resonance imaging. The contributions included time-continuous Transformer blocks and a motion constraint to ensure temporal continuity. As the authors did not provide a response, we have only read the referee report. Feedback from the reviewers included insufficient evaluation (only on ACDC) and unclear Figures as well as incremental work as many of its claimed contributions can be found in existing literature. Results were also modest and did not have much improvement despite the proposed contributions. The paper is very specific to cine MRI and is more suitable for conferences like MICCAI.

**Additional Comments On Reviewer Discussion:**

Feedback from the reviewers included insufficient evaluation (only on ACDC) and unclear Figures as well as incremental work as many of its claimed contributions can be found in existing literature. Results were also modest and did not have much improvement despite the proposed contributions. The paper is very specific to cine MRI and is more suitable for conferences like MICCAI.

The authors did not provide a response to the referee reports.

---

### Decision · Program_Chairs · 2025-01-22

Reject